# Migraine and risk of premature myocardial infarction and stroke among men and women: A Danish population-based cohort study

Cecilia Hvitfeldt Fuglsang[1]*, Lars Pedersen[1], Morten Schmidt[1,2], Jan P. Vandenbroucke[1,3,4], Hans Erik Bøtker[2,5], Henrik Toft Sørensen[1]

1 Department of Clinical Epidemiology, Aarhus University and Aarhus University Hospital, Aarhus, Denmark, 2 Department of Cardiology, Aarhus University Hospital, Aarhus, Denmark, 3 Leiden University Medical Center, Leiden, the Netherlands, 4 London School of Hygiene and Tropical Medicine, University of London, London, United Kingdom, 5 Faculty of Health, Aarhus University, Aarhus, Denmark

* chfn@clin.au.dk

**Data Availability Statement:** The individual-level data used for this study are not publicly available, but can be obtained by application to Statistics Denmark (https://www.dst.dk/en/TilSalg/Forskningsservice/Dataadgang).

## Abstract

### Background

Migraine carries risk of myocardial infarction (MI) and stroke. The risk of premature MI (i.e., among young adults) and stroke differs between men and women; previous studies indicate that migraine is mainly associated with an increased risk of stroke among young women. The aim of this study was to examine impact of migraine on the risk of premature (age ≤60 years) MI and ischemic/hemorrhagic stroke among men and women.

### Methods and findings

Using Danish medical registries, we conducted a nationwide population-based cohort study (1996 to 2018). Redeemed prescriptions for migraine-specific medication were used to identify women with migraine ($n = 179,680$) and men with migraine ($n = 40,757$). These individuals were matched on sex, index year, and birth year 1:5 with a random sample of the general population who did not use migraine-specific medication. All individuals were required to be between 18 and 60 years old. Median age was 41.5 years for women and 40.3 years for men.

The main outcome measures to assess impact of migraine were absolute risk differences (RDs) and hazard ratios (HRs) with 95% confidence intervals (CIs) of premature MI, ischemic, and hemorrhagic stroke, comparing individuals with migraine to migraine-free individuals of the same sex. HRs were adjusted for age, index year, and comorbidities. The RD of premature MI for those with migraine versus no migraine was 0.3% (95% CI [0.2%, 0.4%]; $p < 0.001$) for women and 0.3% (95% CI [−0.1%, 0.6%]; $p = 0.061$) for men. The adjusted HR was 1.22 (95% CI [1.14, 1.31]; $p < 0.001$) for women and 1.07 (95% CI [0.97, 1.17]; $p = 0.164$) for men. The RD of premature ischemic stroke for migraine versus no migraine was 0.3% (95% CI [0.2%, 0.4%]; $p < 0.001$) for women and 0.5% (95% CI [0.1%, 0.8%]; $p <$

**Funding:** CHF is supported by Aarhus University. The funding source had no role in the study design; data collection, analysis, or interpretation; writing the report; or decision to submit the manuscript for publication.

**Competing interests:** CHF owns stock in Novo Nordisk. MS is supported by the Novo Nordisk Foundation (grant NNF19OC0054908). The Department of Clinical Epidemiology, Aarhus University, receives funding for other studies in the form of institutional research grants to (and administered by) Aarhus University. None of these studies has any relation to the present study.

**Abbreviations:** AR, absolute risk; ATC, anatomical therapeutic chemical; CCI, Charlson Comorbidity Index; CI, confidence interval; COPD, chronic obstructive pulmonary disease; CPR, Civil Personal Register; DNPR, Danish National Patient Registry; GP, general practitioner; HR, hazard ratio; IDA, Integrated Database for Labor Market Research; MI, myocardial infarction; NPR, National Prescription Registry; NSAID, nonsteroidal anti-inflammatory drug; PPV, positive predictive value; RD, risk difference; VTE, venous thromboembolism.

0.001) for men. The adjusted HR was 1.21 (95% CI [1.13, 1.30]; $p < 0.001$) for women and 1.23 (95% CI [1.10, 1.38]; $p < 0.001$) for men. The RD of premature hemorrhagic stroke for migraine versus no migraine was 0.1% (95% CI [0.0%, 0.2%]; $p = 0.011$) for women and −0.1% (95% CI [−0.3%, 0.0%]; $p = 0.176$) for men. The adjusted HR was 1.13 (95% CI [1.02, 1.24]; $p = 0.014$) for women and 0.85 (95% CI [0.69, 1.05]; $p = 0.131$) for men.

The main limitation of this study was the risk of misclassification of migraine, which could lead to underestimation of the impact of migraine on each outcome.

## Conclusions

In this study, we observed that migraine was associated with similarly increased risk of premature ischemic stroke among men and women. For premature MI and hemorrhagic stroke, there may be an increased risk associated with migraine only among women.

### Author summary

#### Why was this study done?

- Migraine is associated with an increased risk of myocardial infarction (MI) and stroke.
- Previous studies show that the risk of ischemic stroke among individuals with migraine is increased mainly among young women.
- Whether the increased risk of MI and hemorrhagic stroke among persons with migraine also mainly pertains to women is unknown.

#### What did the researchers do and find?

- The researchers used Danish national registries to identify women and men with and without migraine living in Denmark during the years 1996 to 2018.
- The risk of having an MI or stroke at age 60 or younger was examined for individuals with and without migraine, separately for men and women.
- Migraine was associated with similarly increased risk of ischemic stroke for men and women aged 60 or younger. Migraine may be associated with increased risk of MI and hemorrhagic stroke only in women.

#### What do these findings mean?

- Unlike previous studies, the researchers found that the risk of ischemic stroke in individuals with migraine does not appear to differ substantially between men and women.
- Migraine may be associated with an increased risk of MI and hemorrhagic stroke only among women. However, absolute risks were low.

- Given that migraine is much more prevalent in women, presence of migraine likely represents a greater societal migraine-associated disease burden among women.

## Introduction

Migraine is a common headache disorder with an age-standardized prevalence of 14% [1]. In recent years, migraine has emerged as an established risk factor for myocardial infarction (MI) and stroke. A 2018 meta-analysis demonstrated an approximately 20% increased hazard of MI and an approximately 40% increased hazard of stroke in persons with migraine, compared to those without migraine [2]. As the peak prevalence of migraine occurs at ages 35 to 39 [1], it particularly affects young adults. Since MI and stroke can lead to lifelong disabilities or even death, and prophylaxis using statins and aspirin is available, it is vital to identify persons at increased risk.

As migraine predominantly affects women [1], its impact on risk of MI and stroke may differ for men versus women. Indeed, studies have suggested that impact of migraine on risk of premature (i.e., among young adults) ischemic stroke primarily or exclusively exists for women [3–5]. For instance, a 2017 cohort study on 119,000 individuals with migraine demonstrated that women aged ≤45 years with migraine had an almost 3.5 times higher hazard of ischemic stroke compared to women without migraine. Within the same age group, the hazard was only 1.5 times higher for men with migraine compared to men without migraine [3]. Less is known about whether impact of migraine on risk of premature MI and hemorrhagic stroke differs according to sex. A cohort study from 1995 reporting relative risks of MI suggested that impact of migraine may be greatest in women below 40 years [6]. A South Korean cohort study examined impact of migraine on risk of hemorrhagic stroke and found no impact of migraine on risk of hemorrhagic stroke either among young women or among young men [4]. However, estimates were imprecise in both studies [4,6]. In reviewing the literature, we have found no other studies examining whether impact of migraine on risk premature MI or hemorrhagic stroke differs according to sex. Additionally, in a review article from 2020, Tietjen and Maly suggested that previous studies examining risk of ischemic stroke in young men may have been underpowered due to relative few men with migraine [7].

Therefore, the aim of this study was to examine the impact of migraine on risks of premature MI, ischemic stroke, and hemorrhagic stroke among men and women. We defined premature as an event occurring at or before age 60.

## Methods

This study is reported as per the Strengthening the Reporting of Observational Studies in Epidemiology (STROBE) guideline (S1 STROBE Checklist).

### Setting and study design

The Danish tax-funded healthcare system provides the entire population with access to both primary and secondary healthcare services free of charge [8]. As part of the primary care system, general practitioners (GPs) diagnose, monitor, and treat a variety of diseases. Aside from emergencies, GPs are the first point of contact in the healthcare system and are responsible for referring patients to specialists in private practice, hospital-based outpatient specialty clinics,

and inpatient hospital care when necessary. The costs of prescription drugs are partially reimbursed through the national healthcare system [8].

A unique registration number, the Civil Personal Register (CPR) number, is assigned to all Danish citizens and legal residents at birth or immigration [9]. Contacts with the Danish healthcare system are recorded together with patients' CPR number in national registries. The CPR number thus allows researchers to link data from administrative and medical registries at the individual level [8]. We conducted a nationwide population-based cohort study using Danish registries during the period 1 January 1996 to 31 December 2018.

## Study cohorts

In Denmark, the majority of individuals with migraine are treated by GPs. Individuals may be referred to a hospital in cases of severe migraine or diagnostic uncertainty. While hospital diagnoses are available in the Danish registries, diagnoses given solely by GPs are not. We therefore used records of redeemed migraine medications (including triptans and ergotamine) identified from the Danish National Prescription Registry (NPR) [10] to identify a cohort of males with migraine and a cohort of females with migraine. All redeemed prescriptions issued by GPs, specialists in private practice, and hospitals (treatments administered during admissions not included) have been recorded in the NPR since 1995. Thus, we assumed that this approach would allow us to identify a large and representative sample of individuals with migraine. At redemption, the patient's CPR number, the identifier for the prescriber and dispending pharmacy, and dispensing information are recorded. This information includes the anatomical therapeutic chemical (ATC) classification system code, dispensing date, product name, drug strength and formulation, and number of defined daily doses per package [10].

All men and women aged 18 to 60 years with at least 2 redeemed prescriptions for migraine medication recorded in the NPR were included in one of the 2 sex-specific migraine cohorts (ATC codes in Table A in S4 Text). At least 2 prescriptions were required to increase accuracy of the migraine diagnosis. The index date was defined as redemption date of the second prescription. Individuals were excluded if they had 2 or more prescriptions for migraine medication prior to the study period or a diagnosis of MI or stroke prior to or on the index date. Given the importance of the selection in the study cohort, we also did a sensitivity analysis using migraine diagnoses codes for enrollment in the study; see "Sensitivity analysis" below.

Female and male general population comparison cohorts were constructed using the Danish Civil Registration System [9]. For each individual with migraine, we sampled 5 individuals from the general population matched on sex, birth year, and calendar year. These individuals were assigned the index date corresponding to that of their matched migraine individual. Individuals were eligible for these cohorts if they were alive on the index date, had no previous diagnosis of MI or stroke, and had not redeemed 2 or more prescriptions for migraine medications. If a member of one of the general population cohorts redeemed a second prescription for a migraine drug during follow-up, the individual joined the migraine cohort. However, to avoid informative censoring, the person was not censored from the original cohort.

## Premature MI, ischemic stroke, and hemorrhagic stroke

Cardiovascular events examined were MI, ischemic stroke, and hemorrhagic stroke occurring at age 60 or younger and recorded as primary or secondary discharge diagnoses in the Danish National Patient Registry (DNPR) [11]. We defined these events as premature. Admissions to Danish hospitals have been registered in the DNPR since 1977; emergency room and hospital-based outpatient specialist contacts were added in 1995. Records of each admission or contact include the patient's CPR number, dates of admission and discharge or contact, one primary

diagnosis, and up to 19 secondary diagnoses. Diagnoses were recorded according to the World Health Organization's *International Classification of Diseases*, *Eighth Revision* (ICD-8) until 1994 and then according to the *Tenth Revision* (ICD-10) [11]. To ensure high validity of diagnoses, only inpatient codes were used (ICD codes in Table A in S4 Text). Due to their different pathogenesis, we chose to examine ischemic and hemorrhagic stroke as separate outcomes.

## Covariates

The Integrated Database for Labor Market Research (IDA) [12], established in 1980, contains data on labor market participation for all Danish citizens. This is recorded on a yearly basis together with individuals' CPR number [12]. IDA data on highest level of education achieved and income were used to describe socioeconomic position. For each individual, income was calculated as average income during the 5 years prior to the index year. Income was then categorized for men and women separately as low, medium, high, or very high, based on income quartiles during the year prior to the index year.

We used inpatient, hospital-based outpatient specialty clinic, and emergency room diagnoses recorded in the DNPR to identify comorbidities for each individual in the study prior to or on the index date. The following comorbidities were included: hypertension, atrial fibrillation or flutter, thyroid disease, venous thromboembolism (VTE), valvular heart disease, obesity, alcohol-related disease, mood disorders, and chronic obstructive pulmonary disease (COPD). Furthermore, we identified diseases included in the Charlson Comorbidity Index (CCI) (excluding cerebrovascular disease and MI) [13] and assessed the burden of comorbidity using the CCI score to categorize all individuals into one of 3 CCI categories: low = CCI score of 0 (no comorbidities), medium = CCI score of 1 to 2, or high = CCI score of 3 or more.

For each individual in the study, the NPR was used to identify ever-redeemed prescriptions for beta-blockers, angiotensin-converting enzyme inhibitors or angiotensin II antagonists, diuretics, vitamin K antagonists, direct oral anticoagulants, platelet inhibitors (including clopidogrel and low-dose aspirin), combined oral contraceptives, and systemic progesterone-only contraceptives prior to or on the index date. Additionally, we identified prescriptions for nonsteroidal anti-inflammatory drugs (NSAIDs), which may be used as a migraine abortive treatment. Although these drugs can be purchased at a reduced price if prescribed, they are also available over the counter. No individual-level records exist of over-the-counter purchases. Finally, we identified hyperlipidemia either as an existing diagnosis of hyperlipidemia in the DNPR or as a redeemed prescription for a lipid-lowering drug (ICD and ATC codes in Tables B and C in S4 Text).

## Statistical analyses

We described the study cohorts according to age on the index date, calendar period, socioeconomic position, CCI category, comorbidities, and redeemed prescription drugs. Cohort members were followed from their index dates (date of second redeemed prescription for individuals with migraine) until date of cardiovascular outcome, emigration, 61st birthday, death, or end of follow-up (31 December 2018), whichever came first. To assess the entire impact of migraine on each outcome, we conducted 3 separate analysis—one for each cardiovascular outcome (MI, ischemic stroke, or hemorrhagic stroke). Thus, in the analysis examining impact of migraine on ischemic stroke, we ignored whether or not an MI or hemorrhagic stroke occurred during follow-up and vice versa. S1 Fig illustrates windows for exposure, exclusion, and covariate assessment and follow-up based on template by Schneeweiss and colleagues [14].

The original purpose of this study, as stated in the prespecified protocol (S1 Text), was to examine the interaction between sex and migraine on the risk of premature MI and stroke. However, an interaction analysis demands 2 exposures that are or are not present (with 1 group having none, 2 groups having either, and 1 group having both). It was not clear to us whether men or women would be the "exposed" sex. We therefore changed the aim of the study to examine the impact of migraine on the risk of MI and stroke among women and men separately. The reason for this change was that earlier studies revealed clear sex differences for the risk of ischemic stroke but did not examine sex differences in the risk of MI. We therefore focused on whether the sex differences associated with ischemic stroke also applied to MI and hemorrhagic stroke. The impact of migraine on each outcome for men and women was assessed on the absolute scale, using risk differences (RDs), and on the relative scale, using hazard ratios (HRs). As a measure of absolute risk (AR), cumulative incidences of the outcomes for the follow-up period of 1 to 20 years were calculated for all cohorts using the Aalen–Johansen product-limit estimator, treating death as a competing risk. From these estimates, the RD comparing individuals with migraine to individuals without migraine was calculated for men and women separately. We used Gray's test to compute the $p$-value for comparison of the cumulative incidence curves for men and women separately.

We used Cox regression to calculate crude and adjusted HRs for the comparison of each cohort to women without migraine. We chose to use a Cox model rather than a Fine and Gray model, because the Cox model has been suggested to be more appropriate for etiologic questions [15]. Moreover, we calculated HRs comparing men with migraine to men without migraine. We conducted 2 different sets of adjustments. The first set of adjustments were based on the directed acyclic graphs provided in S2 Text. In this analysis with MI as the outcome, adjustments were made for age (as a continuous variable), calendar period (1996 to 2000, 2001 to 2005, 2006 to 2010, 2011 to 2014, and 2015 to 2018), hypertension, thyroid disease, hyperlipidemia, VTE, obesity, alcohol-related disease, and COPD. Where ischemic stroke was outcome, we adjusted for atrial fibrillation or flutter in addition to the factors adjusted for in the MI analysis. Where hemorrhagic stroke was outcome, we adjusted for age, calendar period, hypertension, alcohol-related disease, COPD, and anticoagulant treatment (vitamin K antagonist, direct oral anticoagulants, or platelet inhibitors). Additional adjustments were made in the second version of the adjusted HRs, and the adjustments were the same for all outcomes. Here, we adjusted for age, calendar period, education, income, CCI score, diabetes, hypertension, atrial fibrillation or flutter, VTE, Raynaud's disease, obesity, thyroid disease, alcohol-related disease, COPD, hyperlipidemia, and ever-redeemed prescriptions of angiotensin-converting enzyme inhibitors/angiotensin II antagonists, diuretics, and anticoagulant treatments. There were no missing data for these analyses. The likelihood ratio test was used to compute $p$-values for the HRs.

Proportionality of hazards was checked using log–log plots (Figs A-C in S3 Text). As the log–log plots showed lack of proportionality for some hazards during the first year, we computed HRs for the follow-up period of years 1 to 20.

## Sensitivity analysis

To investigate the robustness of the main analyses to potential misclassification, we conducted 2 sensitivity analyses. First, we repeated the computation of ARs and HRs of ischemic stroke, including the ICD-10 code for unspecified stroke (I64) in the definition of ischemic stroke (log–log plots in Fig D in S3 Text). An earlier validation study of stroke codes in the DNPR has shown that the majority (57%) of unspecified strokes are ischemic [16].

Second, we repeated all analyses using first-time ICD codes recorded in the DNPR to identify migraine (rather than prescription data). These refer to inpatient or outpatient diagnoses

in hospital (rather than the prescription data from the NPR that refer to all redeemed prescriptions in community pharmacies and, therefore, also include prescriptions initiated by GPs; the NPR was the basis of migraine identification in our main analysis). The hazard rate functions were not completely parallel even looking beyond the first year of follow-up (log–log plots in Figs E and F in S3 Text). Our estimated HRs thus ignore some time-varying effects and show the average impact of migraine [17].

Use of migraine medication changed during the study period. To investigate the importance of this change, we conducted a third sensitivity analysis. In this analysis, we computed HRs of each outcome stratified by index year (1996 to 2006 or 2007 to 2018). To allow for the same potential follow-up, HRs were computed for follow-up of 1 to 10 years.

### Ethics statement

The study was reported to the Danish Data Protection Agency by Aarhus University (record number 2016-051-000001, serial number 603). According to Danish law, no further approvals were required. The funding source had no role in the study design; data collection, analysis, or interpretation; writing the report; or decision to submit the manuscript for publication. Study cohorts were constructed using version SAS version 9.4 (SAS Institute, Cary, NC, USA), and analyses were performed using R software, version 4.2.1.

## Results

### Descriptive data

We identified 179,680 women and 40,757 men with migraine (Table 1). From the general population, we sampled 898,365 women and 203,775 men without migraine. Median age on the index date was 41.5 years for women and 40.3 years for men. A somewhat larger proportion of individuals were identified in the study's earliest calendar period.

Overall, individuals with migraine appeared to have slightly lower socioeconomic position than migraine-free individuals of the same sex (Table 1). Women appeared to have a higher level of education than men. Approximately 86% to 87% of men and women with migraine had none of the comorbidities included in the CCI, whereas approximately 90% to 91% of men and women without migraine were free of these comorbidities on their index date. Among persons of the same sex, prevalence of hypertension, VTE, obesity, thyroid disease, mood disorders, COPD, and hyperlipidemia appeared higher for individuals with migraine than for those without. Prevalence of alcohol-related disease was higher among men with migraine than among men without migraine. Compared to migraine-free individuals of the same sex, a larger proportion of individuals with migraine had redeemed prescriptions for beta-blockers, angiotensin-converting enzyme inhibitors or angiotensin II antagonists, diuretics, platelet inhibitors, and NSAIDs. A larger proportion of women with migraine had redeemed prescriptions for combined oral contraceptives and systemic progesterone-only contraceptives compared to women without migraine. See Table 2 for frequencies of comorbidities and redeemed prescription drugs.

### Risk of premature myocardial infarction, ischemic stroke, and hemorrhagic stroke

For all outcomes, total follow-up was approximately 12.4 million person years and the cohorts were followed for a median of 8.8 years (interquartile range: 4.3 to 4.4, 14.0 years). See Table 3 for number of events and AR estimates for 1 to 20 years of follow-up. The baseline risk of premature MI among individuals without migraine was higher for men than for women: AR 3.1%

**Table 1. Baseline characteristics [n, (%)] of women and men with and without migraine (identified by prescription data), 1996–2018.**

| | Women | | Men | |
|---|---|---|---|---|
| | Without migraine | With migraine | Without migraine | With migraine |
| All | 898,365 | 179,680 | 203,775 | 40,757 |
| Age (median, IQR) | 41.5 [32.3, 49.6] | 41.5 [32.3, 49.6] | 40.3 [32.0, 48.4] | 40.3 [32.0, 48.4] |
| Calendar period | | | | |
| 1996–2000 | 231,900 (25.8) | 46,380 (25.8) | 51,020 (25.0) | 10,204 (25.0) |
| 2001–2005 | 185,590 (20.7) | 37,118 (20.7) | 38,575 (18.9) | 7715 (18.9) |
| 2006–2010 | 190,690 (21.2) | 38,138 (21.2) | 43,420 (21.3) | 8684 (21.3) |
| 2011–2014 | 145,995 (16.3) | 29,199 (16.3) | 34,615 (17.0) | 6923 (17.0) |
| 2015–2018 | 144,190 (16.1) | 28,845 (16.1) | 36,145 (17.7) | 7231 (17.7) |
| Highest level of education[a] | | | | |
| Low | 238,293 (26.5) | 50,650 (28.2) | 53,397 (26.2) | 11,536 (28.3) |
| Medium | 397,022 (44.2) | 82,120 (45.7) | 103,775 (50.9) | 20,646 (50.7) |
| High | 228,784 (25.5) | 41,686 (23.2) | 37,607 (18.5) | 7180 (17.6) |
| Unknown | 34,266 (3.8) | 5224 (2.9) | 8996 (4.4) | 1395 (3.4) |
| Level of income[b] | | | | |
| Low | 259,806 (28.9) | 53,616 (29.8) | 56,186 (27.6) | 11,885 (29.2) |
| Medium | 261,205 (29.1) | 54,810 (30.5) | 59,442 (29.2) | 12,271 (30.1) |
| High | 211,056 (23.5) | 41,000 (22.8) | 46,543 (22.8) | 8816 (21.6) |
| Very high | 161,230 (17.9) | 29,827 (16.6) | 40,393 (19.8) | 7667 (18.8) |
| Missing | 5068 (0.6) | 427 (0.2) | 1211 (0.6) | 118 (0.3) |
| Charlson Comorbidity Index score[c] | | | | |
| Low | 805,932 (89.7) | 154,757 (86.1) | 184,906 (90.7) | 35,373 (86.8) |
| Medium | 82,353 (9.2) | 22,278 (12.4) | 16,378 (8.0) | 4611 (11.3) |
| High | 10,080 (1.1) | 2645 (1.5) | 2491 (1.2) | 773 (1.9) |

[a]Low = primary or lower secondary education; medium = upper secondary or academic professional degree; high = university education at the bachelor degree level or higher.

[b]According to quartiles. Quartiles were calculated separately for women and men.

[c]Charlson Comorbidity Index score: low = 0 (no comorbidities); medium = 1–2; and high = 3 or more.

(95% confidence interval (CI) [2.9%, 3.2%]) versus 1.0% (95% CI [1.0%, 1.1%]). This was also the case for risk of premature ischemic stroke: AR 1.9% (95% CI [1.8%, 2.0%]) versus 1.2% (95% CI [1.1%, 1.2%]). The baseline risk of premature hemorrhagic stroke was nearly the same in men and women without migraine: AR 0.7% (95% CI [0.6%, 0.8%]) versus 0.6% (95% CI [0.5%, 0.6%]) (Fig 1 and Table 3).

The impact of migraine was estimated on the absolute scale as RDs and on the relative scale as HRs. The RD for premature MI among persons with migraine versus no migraine was 0.3% (95% CI [−0.1%, 0.6%]; $p = 0.061$) among men and 0.3% (95% CI [0.2%, 0.4%]; $p < 0.001$) among women. For premature ischemic stroke, the RD was 0.5% (95% CI [0.1%, 0.8%]; $p < 0.001$) for men and 0.3% (95% CI [0.2%, 0.4%]; $p < 0.001$) for women. For premature hemorrhagic stroke as the outcome, the RD was −0.1% (95% CI [−0.3%, 0.0%]; $p = 0.176$) for men and 0.1% (95% CI [0.0%, 0.2%]; $p = 0.011$) for women.

The following reported HRs are those with adjustments based on the directed acyclic graphs (Table 4). The adjusted HR of premature MI among persons with migraine versus no migraine was 1.07 (95% CI [0.97, 1.17]; $p = 0.164$) among men and 1.22 (95% CI [1.14, 1.31]; $p < 0.001$) among women. For premature ischemic stroke, the adjusted HR was 1.23 (95% CI [1.10, 1.38]; $p < 0.001$) for men and 1.21 (95% CI [1.13, 1.30]; $p < 0.001$) for women. For premature

**Table 2. Individual baseline comorbidities and use of prescription drugs [n, (%)] among women and men with and without migraine (identified by prescription data), 1996–2018.**

| | Women | | Men | |
|---|---|---|---|---|
| | **Without migraine** | **With migraine** | **Without migraine** | **With migraine** |
| Diabetes | 10,986 (1.2) | 2001 (1.1) | 2967 (1.5) | 599 (1.5) |
| Hypertension | 18,886 (2.1) | 4434 (2.5) | 3748 (1.8) | 1075 (2.6) |
| Atrial fibrillation or flutter | 2203 (0.2) | 543 (0.3) | 1157 (0.6) | 295 (0.7) |
| VTE[a] | 6637 (0.7) | 1787 (1.0) | 1163 (0.6) | 356 (0.9) |
| Valvular heart disease | 1397 (0.2) | 370 (0.2) | 406 (0.2) | 105 (0.3) |
| Raynaud's disease | 791 (0.1) | 239 (0.1) | 161 (0.1) | 62 (0.2) |
| Obesity | 39,571 (4.4) | 9564 (5.3) | 1930 (0.9) | 593 (1.5) |
| Thyroid disease | 32,445 (3.6) | 7929 (4.4) | 1123 (0.6) | 311 (0.8) |
| Alcohol-related disease | 18,143 (2.0) | 3737 (2.1) | 9902 (4.9) | 2404 (5.9) |
| Mood disorder | 8065 (0.9) | 2715 (1.5) | 872 (0.4) | 434 (1.1) |
| COPD[b] | 7185 (0.8) | 2031 (1.1) | 1568 (0.8) | 510 (1.3) |
| Hyperlipidemia[c] | 24,145 (2.7) | 5988 (3.3) | 7327 (3.6) | 1947 (4.8) |
| Beta-blockers | 71,045 (7.9) | 26,626 (14.8) | 9137 (4.5) | 5159 (12.7) |
| ACE inhibitor/AT II antagonist[d] | 40,625 (4.5) | 10,152 (5.7) | 9580 (4.7) | 2819 (6.9) |
| Diuretics | 89,243 (9.9) | 27,103 (15.1) | 7036 (3.5) | 2021 (5.0) |
| Vitamin K antagonist | 5069 (0.6) | 1305 (0.7) | 1404 (0.7) | 351 (0.9) |
| DOAC[e] | 742 (0.1) | 234 (0.1) | 243 (0.1) | 96 (0.2) |
| Platelet inhibitor[f] | 16,077 (1.8) | 4432 (2.5) | 4162 (2.0) | 1261 (3.1) |
| NSAID[g] | 514,219 (57.2) | 137,748 (76.7) | 105,483 (51.8) | 29,278 (71.8) |
| Combined oral contraceptives | 463,338 (51.6) | 103,814 (57.8) | - | - |
| Systemic progesterone-only contraceptives | 59,753 (6.7) | 17,029 (9.5) | - | - |

[a]VTE, venous thromboembolism.

[b]COPD, chronic obstructive pulmonary disease.

[c]Diagnosis or ever-redeemed prescription.

[d]ACE inhibitor/AT II-antagonist, angiotensin-converting enzyme inhibitor/angiotensin II antagonist.

[e]Direct oral anticoagulants.

[f]Platelet inhibitors including clopidogrel and low-dose aspirin.

[g]Nonsteroidal anti-inflammatory drugs.

hemorrhagic stroke, the adjusted HR was 0.85 (95% CI [0.69, 1.05]; $p = 0.131$) for men and 1.13 (95% CI [1.02, 1.24]; $p = 0.014$) for women. The HR with additional adjustments showed similar results (Table 4).

## Sensitivity analyses

Risk estimates of ischemic stroke (including the code for unspecified stroke) are available in S1 Table. The RDs and HRs comparing individuals with migraine to individuals without migraine were similar to the main analysis.

Defining migraine by ICD codes rather than by redeemed medications, we identified 25,274 women with migraine and 7,397 men with migraine (Table A in S5 Text). The analyses where MI or ischemic stroke was outcome yielded similar results to the main analysis, albeit that the impact of migraine appeared greater for both women and men. For premature hemorrhagic stroke, the RD comparing migraine versus no migraine was −0.1% (95% CI [−0.6%, 0.4%]; $p = 0.403$) for men and 0.3% (95% CI [0.0%, 0.7%]; $p = 0.011$) for women. The adjusted

**Table 3. Number of events, ARs, and RDs for 1–20 years of follow-up for premature MI, ischemic stroke, and hemorrhagic stroke for women and men with and without migraine (identified by prescription data).** *P* values reflect Gray's test for RDs.

|  | Events n | AR % (95% CI) | RD within sex % (95% CI) |
|---|---|---|---|
| **MI** | | | |
| Women without migraine | 3,733 | 1.0 (1.0, 1.1) | - |
| Women with migraine | 929 | 1.3 (1.2, 1.4) | 0.3 (0.2, 0.4); *p* < 0.001 |
| Men without migraine | 2,520 | 3.1 (2.9, 3.2) | - |
| Men with migraine | 548 | 3.3 (3.0, 3.7) | 0.3 (−0.1, 0.6); *p* = 0.061 |
| **Ischemic stroke** | | | |
| Women without migraine | 4,122 | 1.2 (1.1, 1.2) | - |
| Women with migraine | 1,016 | 1.5 (1.3, 1.6) | 0.3 (0.2, 0.4); *p* < 0.001 |
| Men without migraine | 1,448 | 1.9 (1.8, 2.0) | - |
| Men with migraine | 364 | 2.3 (2.0, 2.7) | 0.5 (0.1, 0.8); *p* < 0.001 |
| **Hemorrhagic stroke** | | | |
| Women without migraine | 2,244 | 0.6 (0.5, 0.6) | - |
| Women with migraine | 510 | 0.7 (0.6, 0.8) | 0.1 (0.0, 0.2); *p* = 0.011 |
| Men without migraine | 581 | 0.7 (0.6, 0.8) | - |
| Men with migraine | 101 | 0.6 (0.4, 0.7) | −0.1 (−0.3, 0.0); *p* = 0.176 |

AR, absolute risk; CI, confidence interval; MI, myocardial infarction; RD, risk difference.

HR was 1.16 (95% CI [0.73, 1.83]; *p* = 0.534) among men and 1.40 (95% CI: 1.07, 1.84]; *p* = 0.015) among women. Risk estimates are presented in Table B and Fig A in S5 Text.

When results were stratified by index year (1996 to 2006 or 2007 to 2018), we found a similar impact of migraine on risk of MI for women, although slightly attenuated, for both time periods. For men, we only saw an association with MI in the more recent time period (2007 to 2018). The HRs of ischemic stroke were similar to those of the main analyses for both men and women for both time periods. For premature hemorrhagic stroke, it may be there was an impact of migraine for women identified in 1996 to 2006. However, estimates for this outcome were imprecise. The HRs are available in S2 Table.

## Discussion

In this nationwide population-based cohort study, persons with migraine had an increased risk of ischemic stroke compared to the general population, which was similar for men and women. As baseline risk of premature MI was substantially higher in men than in women and impact of migraine on the absolute scale on risk of this outcome was similar for men and women, the impact on the relative scale appeared larger for women and may potentially only exist for women. For premature hemorrhagic stroke, baseline risks were similar for men and women. We observed a weak association (if any) between migraine and risk of premature hemorrhagic stroke among women, while an influence of migraine on risk of premature hemorrhagic stroke seemed absent among men.

To the best of our knowledge, only Sternfeld and colleagues's study has reported on the impact of migraine on risk of MI among young adults separately for men and women [6]. This study examined relative risk in 2 cohorts. For men with migraine under 40 years of age, the relative risk was 0.3 (95% CI [0.1, 2.4]) in one cohort and 0.6 (95% CI [0.1, 4.4]) in the other cohort. For men with migraine between 40 and 60 years of age, the relative risk was 0.8 (95% CI [0.5, 1.5]) in one cohort and 1.4 (95% CI [0.8, 2.5]) in the other cohort. Risk estimates appeared greatest for women with migraine below 40 years of age, for whom the relative risk

Absolute risk

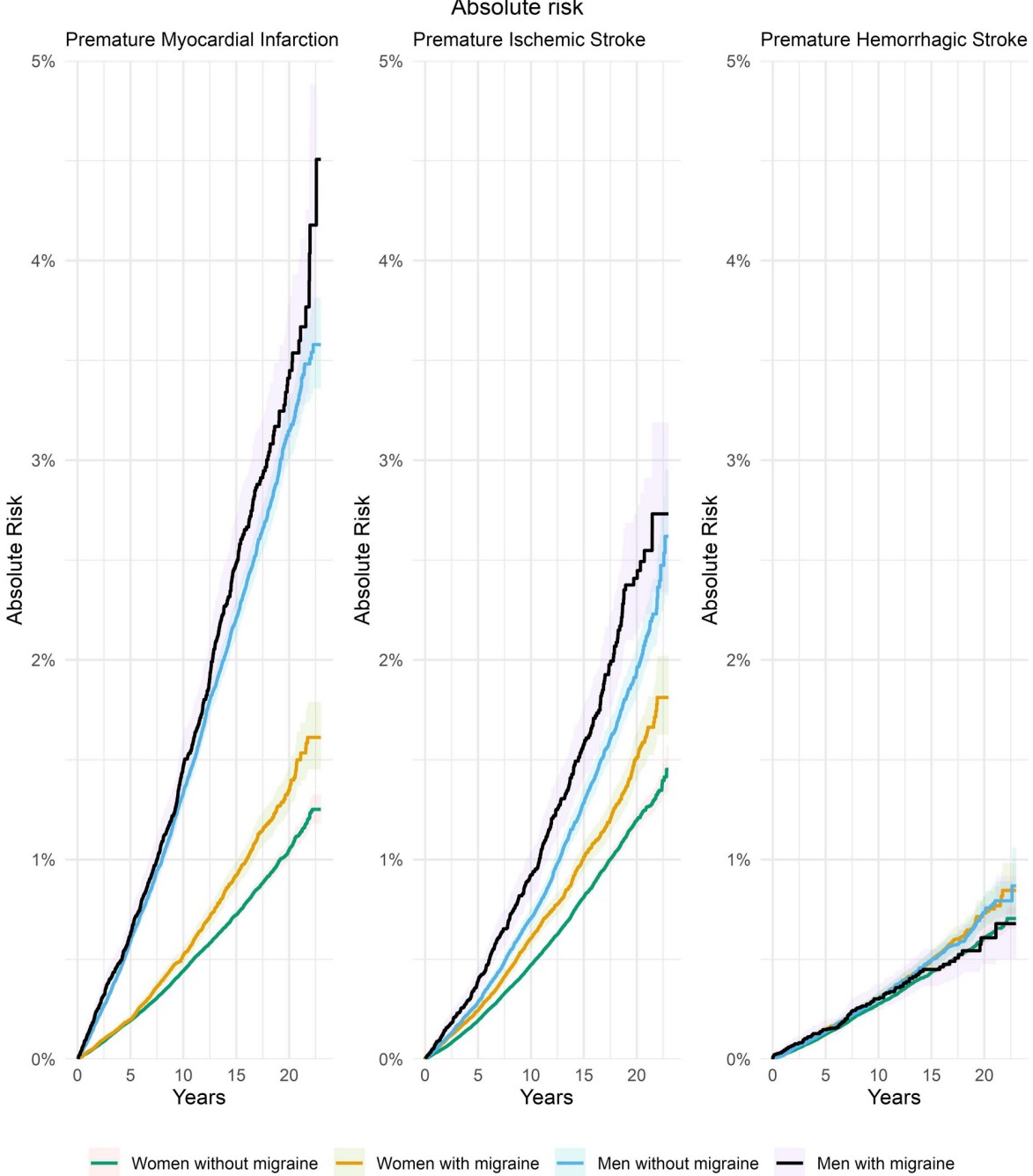

**Fig 1. AR of premature MI, ischemic stroke, and hemorrhagic stroke among women with and without migraine and among men with and without migraine (migraine identified by prescriptions).** The 95% CIs illustrated by the shadowed areas. Difference in impact of migraine was evaluated for men and women separately using Gray's test for the entire follow-up period. For MI and ischemic stroke, the *p*-value was <0.001 for both women and men. For hemorrhagic stroke, the *p*-value was 0.001 for women and 0.517 for men. AR, absolute risk; CI, confidence interval; MI, myocardial infarction.

was 1.5 (95% CI [0.7, 3.5]) in one cohort and 2.1 (95% CI [0.5, 9.5]) in the other cohort. The relative risk for women with migraine aged 40 to 60 years was 1.0 (95% CI [0.6, 1.7]) in one cohort and 1.6 (95% CI [1.0, 2.7]) in the other cohort. Although the impact of migraine appeared greater among women than men, the study's estimates were imprecise [6]. While our

**Table 4. Crude and adjusted HRs for 1–20 years of follow-up for premature MI, ischemic stroke, and hemorrhagic stroke (migraine identified by prescription data).** *P* values reflect likelihood ratio tests for HRs.

| | Crude HR (95% CI) | Crude HR within sex (95% CI) | Adjusted* HR (95% CI) | Adjusted* HR within sex (95% CI) | HR with additional adjustments† (95% CI) | HR within sex with additional adjustments† (95% CI) |
|---|---|---|---|---|---|---|
| **Myocardial infarction** | | | | | | |
| Women without migraine | 1.00 (Ref) | 1.00 (Ref) | 1.00 (Ref) | 1.00 (Ref) | 1.00 (Ref) | 1.00 (Ref) |
| Women with migraine | 1.24 (1.15, 1.33); *p* < 0.001 | 1.24 (1.15, 1.33); *p* < 0.001 | 1.22 (1.14, 1.31); *p* < 0.001 | 1.22 (1.14, 1.31); *p* < 0.001 | 1.18 (1.09, 1.27); *p* < 0.001 | 1.18 (1.09, 1.27); *p* < 0.001 |
| Men without migraine | 3.07 (2.92, 3.23); *p* < 0.001 | 1.00 (Ref) | 3.20 (3.04, 3.37); *p* < 0.001 | 1.00 (Ref) | 3.27 (3.11, 3.45); *p* < 0.001 | 1.00 (Ref) |
| Men with migraine | 3.34 (3.05, 3.65); *p* < 0.001 | 1.09 (0.99, 1.19); *p* = 0.08 | 3.39 (3.09, 3.71); *p* < 0.001 | 1.07 (0.97, 1.17); *p* = 0.164 | 3.43 (3.13, 3.76); *p* < 0.001 | 1.06 (0.96, 1.16); *p* = 0.230 |
| **Ischemic stroke** | | | | | | |
| Women without migraine | 1.00 (Ref) | 1.00 (Ref) | 1.00 (Ref) | 1.00 (Ref) | 1.00 (Ref) | 1.00 (Ref) |
| Women with migraine | 1.23 (1.15, 1.31); *p* < 0.001 | 1.23 (1.15, 1.31); *p* < 0.001 | 1.21 (1.13, 1.30); *p* < 0.001 | 1.21 (1.13, 1.30); *p* < 0.001 | 1.18 (1.10, 1.26); *p* < 0.001 | 1.18 (1.10, 1.26); *p* < 0.001 |
| Men without migraine | 1.59 (1.50, 1.69); *p* < 0.001 | 1.00 (Ref) | 1.60 (1.50, 1.70); *p* < 0.001 | 1.00 (Ref) | 1.63 (1.53, 1.73); *p* < 0.001 | 1.00 (Ref) |
| Men with migraine | 2.00 (1.80, 2.23); *p* < 0.001 | 1.26 (1.12, 1.41); *p* < 0.001 | 1.94 (1.74, 2.16); *p* < 0.001 | 1.23 (1.10, 1.38); *p* < 0.001 | 1.95 (1.75, 2.18); *p* < 0.001 | 1.22 (1.09, 1.37); *p* < 0.001 |
| **Hemorrhagic stroke** | | | | | | |
| Women without migraine | 1.00 (Ref) | 1.00 (Ref) | 1.00 (Ref) | 1.00 (Ref) | 1.00 (Ref) | 1.00 (Ref) |
| Women with migraine | 1.13 (1.03, 1.24); *p* = 0.012 | 1.13 (1.03, 1.24); *p* = 0.012 | 1.13 (1.02, 1.24); *p* = 0.014 | 1.13 (1.02, 1.24); *p* = 0.014 | 1.09 (0.99, 1.21); *p* = 0.072 | 1.09 (0.99, 1.21); *p* = 0.072 |
| Men without migraine | 1.17 (1.07, 1.28); *p* < 0.001 | 1.00 (Ref) | 1.15 (1.05, 1.26); *p* = 0.003 | 1.00 (Ref) | 1.16 (1.05, 1.27); *p* = 0.003 | 1.00 (Ref) |
| Men with migraine | 1.02 (0.83, 1.24); *p* = 0.865 | 0.87 (0.70, 1.07); *p* = 0.189 | 0.98 (0.80, 1.20); *p* = 0.842 | 0.85 (0.69, 1.05); *p* = 0.131 | 1.00 (0.82, 1.22); *p* = 0.991 | 0.85 (0.69, 1.06); *p* = 0.143 |

*For MI: adjusted for age, calendar period, hypertension, thyroid disease, hyperlipidemia, VTE, obesity, alcohol-related disease, and COPD.

For ischemic stroke: adjusted for age, calendar period, hypertension, thyroid disease, hyperlipidemia, VTE, obesity, alcohol-related disease, COPD and atrial fibrillation/flutter.

For hemorrhagic stroke: adjusted for age, calendar period, hypertension, alcohol-related disease, COPD, and anticoagulant treatment.

†All outcomes adjusted for education, income, CCI score, diabetes, hypertension, atrial fibrillation/flutter, VTE, Raynaud's disease, obesity, thyroid disease, alcohol-related disease, COPD, hyperlipidemia, and ever-redeemed prescriptions of angiotensin-converting enzyme inhibitors/angiotensin II antagonists, diuretics, and anticoagulant treatments.

CCI, Charlson Comorbidity Index; CI, confidence interval; COPD, chronic obstructive pulmonary disease; HR, hazard ratio; MI, myocardial infarction; VTE, venous thromboembolism.

study's estimates had greater precision, they showed similarly that the impact of migraine on risk of MI on a relative scale may be slightly greater for women.

Regarding the impact of migraine on ischemic stroke, earlier studies indicated that migraine is a more important risk factor for young women than for young men. Peng and colleagues [3] reported an adjusted HR of 3.44 (95% CI [2.0, 5.38]) for ischemic stroke among women with migraine aged ≤45 years, compared to women without headache disorders. The adjusted HR was 1.54 (95% CI [0.96, 2.48]) for ischemic stroke among men with migraine ≤45 years, compared to men without headache disorders. Lee and colleagues [4] reported HRs of ischemic stroke of 2.31 (95% CI [1.31, 3.82]) and 1.32 (95% CI [1.08, 1.61]) for women with

migraine compared to migraine-free women aged 20 to 39 and 40 to 59, respectively. For men, the HRs were 1.72 (95% CI [0.89, 3.32]) among persons with migraine aged 20 to 39 years, and 0.89 (95% CI [0.68, 1.17]) among those aged 40 to 59. The contrast between ischemic stroke risk in young men versus young women in these studies thus was somewhat larger than our findings suggest. Lee and colleagues [4] also investigated migraine-associated risk of hemorrhagic stroke stratified on age and sex. Among women, the HR was 1.05 (95% CI [0.59, 1.84]) within the 20- to 39-year age group and 1.12 (95% CI [0.87, 1.44]) within the 40- to 59-year age group. For men, the HRs were 1.37 (95% CI [0.73, 2.59]) among those aged 20 to 39 years and 1.09 (95% CI [0.74, 1.61]) among those aged 40 to 59 years. The investigators concluded that for all migraine groups, risk of hemorrhagic stroke was similar to that for controls. The HRs that they reported for young adults resulted from stratification by age at index year and therefore do not indicate risk of premature stroke or MI. We thus cannot make direct comparisons with our study, which only included events if they occurred at age 60 or younger.

The reason for the association between migraine and cardiovascular disease remains unclear, although several explanations have been suggested [18,19]. The cortical spreading depressions associated with migraine with aura could dispose to ischemia of the brain [19]. Migraine also has been associated with increased prevalence of patent foramen ovale [20] and increased levels of von Willebrand factor [21], which might contribute to increased risk of ischemic stroke. A possible association between migraine and coronary vasospasms has been suggested, leading to the theory that migraine may be a generalized disorder affecting vascular tone both of the brain and the coronary arteries [22,23]. Still, research on such an association is limited and the theory has been rebutted [24,25] and may also be related to side effects by medication against migraine, for instance, sumatriptan [26]. The association between migraine and cardiovascular disease also has been suggested to be genetic or due to cardiovascular risk factors associated with migraine, such as hypertension, obesity, smoking, and NSAID use [19]. At the same time, migraine headaches could be associated with aspirin intake. While aspirin should have a protective effect on risk of MI, it could explain an increased risk of hemorrhagic stroke. However, the consumption of aspirin or other anticoagulants would have to be greater among women compared to men to explain our results. Due to the preponderance of women among persons with migraine and the link between frequency of migraine attacks and hormone fluctuations during the menstrual cycle, estrogen is believed to be involved in migraine pathophysiology [27]. Furthermore, endogenous estrogen is believed to be protective of cardiovascular disease [28]. Thereby, estrogen has been suggested to be the link between migraine and cardiovascular risk among women [29]. However, as our findings suggest the increased risk of at least ischemic stroke also exists among young men, other mechanisms may be at play. Further research is needed to explain the relation between migraine and cardiovascular risk and to confirm or rule out whether the impact of migraine on cardiovascular risk differs for young men and women.

Limitations include possible misclassification of migraine. We used redeemed prescriptions to identify migraine. In Denmark, migraine should be the only indication for the prescriptions we included except for sumatriptan, which may be administered subcutaneously to treat cluster headache [30]. Cluster headache is, however, a rare condition (estimated prevalence of 0.1%; [31]), and we therefore assumed misclassification due to this to be low. Other potential misclassification would most likely be nondifferential and bias towards the null for both women and men. We chose to use the prescription data for the main analyses as this allowed us to include a large sample of individuals with migraine. At the same time, these cohorts would presumably represent a broader spectrum of individuals with migraine compared to those with a hospital-diagnosis (in- or outpatient) who are captured in the DNPR; the latter would most likely represent individuals more severely affected by their migraine. Still, we

examined impact of migraine using diagnostic codes to identify migraine. This yielded similar results, although impact of migraine appeared greater for both men and women. For premature hemorrhagic stroke, impact of migraine on the absolute scale was still greatest for women. Previous studies have shown that the increased cardiovascular risk in individuals with migraine is stronger in those with aura compared to those without but exists for both forms of migraine [32], which was recently shown again in Danish population data [33]. The prescription data did not allow us differentiate between migraine with and without aura. In addition, the paucity of events, despite the follow-up of large numbers of individuals, would prevent meaningful analyses of these subgroups in the diagnosed migraine individuals. Diagnostic codes of the outcomes, recorded in the DNPR, have been validated previously with overall relatively high positive predictive values (PPVs). Validation studies on stroke have reported PPVs of 79.3% to 80.5% [16,34], while the PPV for a first-time MI has been estimated as 97% [35]. Any misclassification of MI is likely to be nondifferential. As a migraine aura can mimic a stroke, a diagnosis of stroke actually could be a migraine attack. This type of bias could create or augment an association between migraine and stroke. Further, despite adjustment of HRs for certain comorbidities, residual confounding could have remained. In addition, unrecorded cardiovascular risk factors such as exercise, alcohol intake, and smoking could be potential confounders.

In our study, we did not find the large differences reported in previous studies as migraine was associated with similarly increased risks of premature ischemic stroke among men and women. The impact of migraine on risk of premature MI may be slightly greater for women or potentially only exist for women. Furthermore, a minor impact of migraine may be present for hemorrhagic stroke among women only. Of note, given that migraine is much more prevalent in women (approximately 179,000 women versus approximately 40,000 men with migraine in this population-based study), the presence of migraine will likely result in a greater migraine-associated disease burden in society for women.

## Supporting information

**S1 STROBE Checklist. STROBE Statement—Checklist of items that should be included in reports of cohort studies.**
(DOCX)

**S1 Fig. Graphical depiction of study design and windows for exclusion, exposure, covariate, and follow-up assessment.** Legend: *Earliest of cardiovascular event, 61st birthday, emigration, death, or administrative censoring (31 December 2018). Based on template by Schneeweiss and colleagues [14].
(DOCX)

**S1 Text. Initial statistical analysis plan.**
(DOCX)

**S2 Text. Directed acyclic graphs illustrating potential confounders for the analyses.** Legend: Purple arrows illustrate potential biasing path and green arrows illustrate causal paths. Light grey covariates indicate that we did not have information on the variable.
(DOCX)

**S3 Text. Log–log plots.**
(DOCX)

**S4 Text. ICD codes and ATC codes for exposure, outcomes, and covariates.**
(DOCX)

**S5 Text. Sensitivity analysis 2: Identifying migraine using diagnosis codes in the DNPR.**
(DOCX)

**S1 Table. Sensitivity analysis 1: Risk of ischemic stroke including ICD code I64.**
(DOCX)

**S2 Table. Sensitivity analysis 3: 1–10 year hazard ratios stratified by index year (1996–2006, 2007–2018).**
(DOCX)

## Author Contributions

**Conceptualization:** Cecilia Hvitfeldt Fuglsang, Lars Pedersen, Morten Schmidt, Henrik Toft Sørensen.

**Data curation:** Lars Pedersen, Henrik Toft Sørensen.

**Formal analysis:** Cecilia Hvitfeldt Fuglsang, Lars Pedersen.

**Funding acquisition:** Cecilia Hvitfeldt Fuglsang, Henrik Toft Sørensen.

**Investigation:** Cecilia Hvitfeldt Fuglsang, Henrik Toft Sørensen.

**Methodology:** Cecilia Hvitfeldt Fuglsang, Jan P. Vandenbroucke, Hans Erik Bøtker, Henrik Toft Sørensen.

**Project administration:** Cecilia Hvitfeldt Fuglsang, Henrik Toft Sørensen.

**Software:** Cecilia Hvitfeldt Fuglsang, Lars Pedersen.

**Supervision:** Lars Pedersen, Morten Schmidt, Jan P. Vandenbroucke, Hans Erik Bøtker, Henrik Toft Sørensen.

**Visualization:** Cecilia Hvitfeldt Fuglsang.

**Writing – original draft:** Cecilia Hvitfeldt Fuglsang.

**Writing – review & editing:** Cecilia Hvitfeldt Fuglsang, Lars Pedersen, Morten Schmidt, Jan P. Vandenbroucke, Hans Erik Bøtker, Henrik Toft Sørensen.

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
