## [Editor Report · Decision Letter 0]

29 Dec 2022

Dear Dr Hvitfeldt Fuglsang, 

Thank you for submitting your manuscript entitled "Impact of migraine on risk of premature myocardial infarction and stroke among men and women: A Danish population-based cohort study" for consideration by PLOS Medicine.

Your manuscript has now been evaluated by the PLOS Medicine editorial staff and I am writing to let you know that we would like to send your submission out for external peer review.

Please re-submit your manuscript within two working days, i.e. by Jan 02 2023 11:59PM.

Kind regards,

Philippa Dodd, MBBS MRCP PhD

PLOS Medicine

---

## [Decision Letter · Decision Letter 1]

13 Feb 2023

Dear Dr. Hvitfeldt Fuglsang,

Thank you very much for submitting your manuscript "Impact of migraine on risk of premature myocardial infarction and stroke among men and women: A Danish population-based cohort study" (PMEDICINE-D-22-03940R1) for consideration at PLOS Medicine. 

[LINK]

In light of these reviews, I am afraid that we will not be able to accept the manuscript for publication in the journal in its current form, but we would like to consider a revised version that addresses the reviewers' and editors' comments. Obviously we cannot make any decision about publication until we have seen the revised manuscript and your response, and we plan to seek re-review by one or more of the reviewers. 

We expect to receive your revised manuscript by Mar 06 2023 11:59PM. Please email us (plosmedicine@plos.org) if you have any questions or concerns.

We look forward to receiving your revised manuscript. 

Sincerely,

Philippa Dodd, MBBS MRCP PhD

PLOS Medicine

plosmedicine.org

GENERAL

Please respond to all editor and reviewer requests as detailed below, in full.

Please include line numbers in your revised version starting at 1 and continuing in sequence thereafter

Please ensure that the study is reported according to the STROBE guideline, and include the completed STROBE checklist as Supporting Information. Please add the following statement, or similar, to the Methods: "This study is reported as per the Strengthening the Reporting of Observational Studies in Epidemiology (STROBE) guideline (S1 Checklist)."

When completing the checklist, please use section and paragraph numbers, rather than page and/or line numbers, as these often change in the event of publication.

ABSTRACT

Please structure your abstract using the PLOS Medicine headings (Background, Methods and Findings, Conclusions). Please combine the Methods and Findings sections into one section, “Methods and findings”.

Abstract Background: 

Please ensure that the final sentence clearly states the study question.

Abstract Methods and Findings: 

Please ensure that all numbers presented in the abstract are present and identical to numbers presented in the main manuscript text.

Please expand the details of your study population – number of participants, age range (and mean) how did you select them? And so on.

Please detail the main outcome measures.

Please quantify the main results p values as well as 95% CIs. Please ensure that you report p as <0.001 or where higher as p=0.002, for example.

Suggest reporting statistical information as follows: “…0.3% (95% CI [-0.1%, 0.6%]; p<0.001 or p=0.002), for clarity and accessibility. Please check and amend throughout for consistency in data reporting.

Please detail factors that are adjusted for in the analyses you refer to

In the last sentence of the Abstract Methods and Findings section, please describe the main limitation(s) of the study's methodology.

Abstract Conclusions:

Please interpret the study based on the results presented in the abstract, emphasizing what is new without overstating your conclusions. The phrase "In this study, we observed ..." may be useful.

Please avoid vague statements such as "these results have major implications for policy/clinical care". Mention only specific implications substantiated by the results.

Please avoid assertions of primacy ("We report for the first time....")

AUTHOR SUMMARY

At this stage, we ask that you include a short, non-technical Author Summary of your research to make findings accessible to a wide audience that includes both scientists and non-scientists. The Author Summary should immediately follow the Abstract in your revised manuscript. This text is subject to editorial change and should be distinct from the scientific abstract. It may be helpful to you to review some examples in papers on our website here: https://journals.plos.org/plosmedicine/

Please see our author guidelines for more information, here: https://journals.plos.org/plosmedicine/s/revising-your-manuscript#loc-author-summary

INTRODUCTION

The introduction is rather brief and lacks some degree of context. Please expand details of the background and importance of the study. See reviewer #2 comments (below) also.

Please also indicate whether your study is novel and how you determined that. 

METHODS and RESULTS

Please add the following statement, or similar, to the Methods: "This study is reported as per the Strengthening the Reporting of Observational Studies in Epidemiology (STROBE) guideline (S1 Checklist)."

Did your study have a prospective protocol or analysis plan? Please state this (either way) early in the Methods section.

For all observational studies, we ask authors to indicate the following in the manuscript text: 

(1) the specific hypotheses you intended to test, 

(2) the analytical methods by which you planned to test them, 

(3) the analyses you actually performed, and 

(4) when reported analyses differ from those that were planned, transparent explanations for differences that affect the reliability of the study's results. If a reported analysis was performed based on an interesting but unanticipated pattern in the data, please be clear that the analysis was data-driven.

Where you report 95% CIs please also report p values. Please report p as p<0.001 and where higher as p=0.002, for example

As above, suggest reporting statistical information as follows: “…0.3% (95% CI [-0.1%, 0.6%]; p<0.001 or p=0.002), for clarity and accessibility. Please check and amend throughout for consistency in data reporting.

Please see methodological reviewer (#3) comments below which we agree with

TABLES

Please include p values where relevant, reported as detailed above.

In the table caption please detail the statistical test used to determine them

There is some inconsistency in how your separate upper and lower confidence limits (semicolons Vs commas). Suggest the use of commas. Please amend throughout where relevant

FIGURES

Please consider using a colour palate which is accessible to those with colour blindness

Figure 1 – does the shadow behind the lines have a meaning/purpose? Please clarify/revise as necessary.

SUPPORTING FILES

Appendix page 4 – suggest removing “log-log plots” above the figure caption

In my version the legends which depict blue and red lines are very small and not easy to read, please revise

As above, suggest the use of commas to separate upper and lower confidence limits. Please amend throughout where relevant

REFERENCES

For in-text reference call-outs, citations should be placed in square brackets and preceding punctuation as for example “…[1-4,7], or [1,5,9].”

In the bibliography please ensure that up to but no more than 6 author names are listed followed by et al., in the event that more than 6 authors contribute to the study.

Please ensure that journal name abbreviations should be those found in the National Center for Biotechnology Information (NCBI) databases.

Please see our website for other reference guidelines https://journals.plos.org/plosmedicine/s/submission-guidelines#loc-references

Comments from the reviewers:

Reviewer #1: This is an interest study investigating the association between impact of migraine on risk of premature myocardial infarction and stroke among men and women in Denmark. There are several strengths of the study: It is built on the Danish registries with diagnoses based on the National Patient Register, it is a nested-case control study with cases matched 1:5 with a random sample of the general Danish population, confounder adjustment is built on DAGS, there are appropriate sensitivity analyses, and the results are presented with both relative and absolute risks. 

However, the following should be considered by the authors before publication: 

Sex differences. In your aim you talk about men and women, corresponding to stratified analyses, but you draw conclusions regarding sex differences in the associations; however, these differences are not tested. So, if you want to examine sex differences please state p-values for the interactions between sex and migraine in all models. Otherwise, you can talk about men and women in your results and conclusions but not about sex differences. 

Study design: This is a nested case-control study - it should be stated in the text under study cohorts and in the abstract. Is birth equal to birthdate? (Page 4 under matching). 

Stroke: You separate the two types of strokes - ischemic stroke and hemorrhagic stroke - however, your results would be easier to understand if you present the two types together in the main analyses and then show the separate models after that. You also need to explain for the reader why you also show the analyses separated by stroke type if you prefer to demonstrate this as well. 

Cox proportional hazards assumptions: You have problems in many of your models with the Cox proportional hazards assumptions. Another way to deal with this is besides excluding the first year of follow up like you did, is to make a stratified piecewise Cox regression, so you investigate different time-periods separately. Particularly in the models where you repeat the analyses using first-time ICD-codes recorded in the DNPR to identify migraine in stead of the prescription date you run into problems - this is an important analysis. Could you fix the problem by a stratified piecewise cox regression and try to combine the two types of stroke? 

Conclusion: Your conclusion should be rewritten. "A slightly increased risk (if any)" is not specific enough in the conclusion. The last part from "despite small differences…" should be moved for instance into the introduction. Your conclusion should sum up clear and concise what you find in this study, and I think a combination of the two types of stroke would make it easier. 

Reviewer #2: Review:

It was interesting to read this study on migraine and risk for myocardial infarction, ischemic and hemorrhagic stroke. It uses large registers both use of migraine medicines and diagnoses from healthcare in a clever way. I believe the method is sound and the results are reasonable. 

My greatest concern is labeling and the fact that almost 180 000 women redeemed migraine medicine but only approx. 25000 women received a migraine diagnose from healthcare. I also didn't understand if a person had more than one of the three diagnoses (MI, IS or ICH) was the persons data then used several times? There are also very much additional data in the appendix, consider if all are really needed (could not several pages of log-log curves be summarized in a sentence what they are showing and then "data not shown"? Are the DAG graphs necessary?). 

Introduction

The main pieces are there, but the words combining them is sort of missing (like on a plate you have the main pieces of food but no sauce). Perhaps a sentence defining the knowledge gap would improve? And why is this study needed if other researchers have done similar (it is needed but argue for it)? I am not fond of this way of writing: "A 2018 meta-analysis…". Should the aim not state the age limit i.e., <60 years? 

Method

How old was the youngest participant? Even if it is very rare, were there any individuals below 18 years?

I enjoyed the use of covariates. However, I was thinking why beta-blockers are used (page 5). It seems as if it was used to identify cardiac disease or hypertension, but it is also sometimes used as prophylactic for migraine.

Page 6 states index-date but does not define this. Is it for the second redeemed prescription of migraine medicine or the first cardiovascular event (MI, IS or ICH)?

During the study period there was a change in use of migraine medicine, at least an increase of triptane use. Did that affect the results?

Results

Page 8, sixth line should not the second "4" have a decimal? (Interquartile range: 4.3-4; 14.0 years) 

Discussion

First paragraph hardly does not mention ischemic stroke.

Table/figures

Why is not Table 1a and 1b separated? They are clearly two different Tables?

Table 1a the heading does not describe how migraine is defined

Table 1b what is "Platelet inhibitor" and why is it separated from Clopidogrel and Aspirin? In Table A2 there is a different set up.

Figure 1 - the grey area of the curves is not defined.

Figure A3 - Why different colors? Does that have a meaning, if so clarify.

Table 2 and 6A is not clearly defined what separates the two.

References

Some references use full names (for instance 5-9) and other uses first names as initials. Why the differences?

Why does the first name in the list of authors have their first name initials before the last name and the rest of the authors have their first name initials after the last name?

Reviewer #3: This is an interesting and well-conducted population-based study on the impact of migraine on risk of premature myocardial infarction and stroke among men and women. However, there are a few major issues needing attention especially with statistical analyses.

1) Covariates in the adjusted Cox models. It says "Covariates were chosen based on the directed acyclic graphs provided in Appendix Figures A1-3". However, this is simply neither enough nor adequate. Confounder/covariate/model selection can not be only based on DAGs. As this is an observational study, we need to minimise the confounding effect as much as possible therefore a full and comprehensice adjustment for the models is needed. Basically all the available variables in the table 1a and 1b needed to be considered in the adjustment, plus drinking, smoking, BMI, depression and anxiety, and substance use, if possible.

2) Competing risk. For the Cox models, as the outcomes are MI and strokes other than all cause mortality, competing risk from deaths (n=?) needs to be addressed in the models, normally by Fine-Gray models. So far, it's not done yet in the paper.

3) It's adequate to use the Aalen-Johansen product-limit estimator, treating death as a competing risk, for cumulative incidences of the outcomes. However, P-values are needed for the statistical differences between the curves (eg. migraine vs non-migraine). This applies to Figure 1 and all other figures in the appendix.

4) Interpretation of the results. It says in the conclusion of the abstract that "Migraine was associated with similarly increased risks of premature MI and ischemic stroke among men and women and, potentially, a minor increased risk of premature hemorrhagic stroke only among women". However, only those adjusted HRs gave conclusive answers, not those AR or RD as they are not adjusted at all. For this type of cohort study, fully-adjusted analyses are absolutely essential. By looking at those adjusted HRs, we can only say migraine is associated with increased risk of MI, ischemic and hemorrihagic stroke mainly for women, not for men (except for ischemic stroke).

[LINK]

---

## [Decision Letter · Decision Letter 2]

21 Apr 2023

Dear Dr. Hvitfeldt Fuglsang,

Thank you very much for re-submitting your manuscript "Impact of migraine on risk of premature myocardial infarction and stroke among men and women: A Danish population-based cohort study" (PMEDICINE-D-22-03940R2) for review by PLOS Medicine.

I have discussed the paper with my colleagues and the academic editor and it was also seen again by 3 reviewers. I am pleased to say that provided the remaining editorial and production issues are dealt with we are planning to accept the paper for publication in the journal.

[LINK]

We look forward to receiving the revised manuscript by Apr 28 2023 11:59PM.   

Sincerely,

Philippa Dodd, MBBS MRCP PhD

PLOS Medicine

plosmedicine.org

Requests from Editors:

GENERAL

Thank you for your very detailed and considered responses to previous editor and reviewer comments. Please see below for further comments which we require you address in full.

* With reference to reviewer #2 comment #3 regarding additional data, the editorial team are in agreement that the information should remain as supporting files as PLOS Medicine does not permit statements such as “data not shown” *

** The editorial team agree that the tables included in your rebuttal need not be included for publication in the main manuscript. They will however be available to review as part of the published peer review **

TITLE

Suggest “Migraine and risk of…” instead

ABSTRACT

Line 37 – “Five times as many individuals…” onwards, would it be simpler to state “These individuals were matched 1:5 with a random sample of the general population…” or similar?

AUTHOR SUMMARY

This largely reads very nicely. However, you state that migraine increases risk of MI and stroke (1st statement) then latterly state that the risk of MI is unknown, presumably you are referring to sex specific risk Vs total (affected) population risk? Please amend/revise for clarity.

Line 80 – sentence beginning “Migraine may be…” suggest making a separate bullet point

TABLES

Table 1 title/caption – suggest, “… [n, (%)]…”

REFERENCES

For in-text reference callouts, please remove spaces between different citations. For example, line 430 should read “…[16,31]…” as opposed to “…[16, 31]…” please check and amend throughout.

SOCIAL MEDIA

To help us extend the reach of your research, if not already done so please provide any Twitter handle(s) that would be appropriate to tag, including your own, your coauthors’, your institution, funder, or lab. Please detail any handles you wish to be included when we tweet this paper, in the manuscript submission form when you re-submit the manuscript.

Comments from Reviewers:

Reviewer #1: The authors have addressed all my comments from the first version of the manuscript or at least answered why they have not changed the manuscript accordingly. 

I can accept the current version and will let the editor decide whether the extra tables should be included in the manuscript. 

Reviewer #2: I think the authors have done an excellent job answering my concerns! Even though my epidemiological skills are not at a level to fully understand the concerns mentioned by Reviewer #3, I feel that I am convinced by the results of the paper.

Only minor things, optional for the authors to change:

1. Line 391-408. It is interesting to discus the possible ways that migraine can cause MI och stroke, but are there theories about the sex difference? Sex is only discussed concerning antithrombotics, not focused on the main topic of the paper. 

2. In "S8 - Sensitivity analysis 2: Identifying migraine using diagnosis codes in the DNPR." the figure for women and men with/without migraine are all straight lines. If printed on black and white it would be easier if some lines are straight, others dotted, lined and so on.

Reviewer #3: Many thanks authors for their great effort to improve the manuscript. All my comments/concerns were comprehensively addressed. I am satisfied with the response and revision. No further issues needing atttention.

[LINK]

---

## [Editor Report · Decision Letter 3]

26 Apr 2023

Dear Dr Hvitfeldt Fuglsang, 

On behalf of my colleagues and the Academic Editor, Dr. Sanjay Basu, I am pleased to inform you that we have agreed to publish your manuscript "Migraine and risk of premature myocardial infarction and stroke among men and women: A Danish population-based cohort study" (PMEDICINE-D-22-03940R3) in PLOS Medicine.

PRESS

Thank you again for submitting to PLOS Medicine, it has been a pleasure handling your manuscript. We look forward to publishing your paper. 

Best wishes,

Pippa 

Philippa Dodd, MBBS MRCP PhD 

PLOS Medicine